# EDITORIAL

## Recognizing excellence in peer review

Kim E. Barrett[1] and Damian M. Bailey[2]

[1]*School of Medicine, University of California Davis, Sacramento, California, USA*
[2]*Neurovascular Research Laboratory, Faculty of Life Sciences and Education, University of South Wales, Pontypridd, UK*

Email: kbarrett@health.ucdavis.edu

The peer review history is available in the Supporting Information section of this article (https://doi.org/10.1113/JP290779#support-information-section).

The world of scientific publishing continues to be beset by challenges, both internal and external. From paper mills to the role of artificial intelligence (AI) to the positives and negatives of open access business models and many more, some days the only thing that seems constant is change. But there is one aspect that, in our view, is unchanging, and that is the benefit of insightful and rigorous, yet respectful, peer review to elevate the quality and impact of the work that fills our pages. Of course, behind these peer reviews there are peer reviewers (working scientists with papers of their own), who contribute altruistically to a mutually beneficial cooperative that raises the value of the literature as a whole. We have celebrated these critical contributors previously in *The Journal of Physiology* (Forsythe, 2017). Much has been written about the limitations of the current peer review system, especially the challenges in finding (and keeping) outstanding reviewers when we all have so many demands on our time and the number of manuscripts in the system continues to grow exponentially. It is now increasingly common to invite more than a dozen potential reviewers, and sometimes many more, to secure two who will commit to providing a report.

In fact, serving as a peer reviewer, albeit time-consuming, carries many benefits. For some, the feeling of simply providing a service to one's colleagues and the field, in the expectation that this service will be repaid, is ample compensation. But there are other advantages. First, it is a privilege to have a front-row seat as new discoveries are reported, refined and, ultimately, published. The typically anonymous reviewers play a pivotal role in shaping the direction of our discipline, often bringing new insights around data interpretation or critical additional experiments that illuminate underlying mechanisms more definitively. Those with some distance from the work at hand are often able to identify issues that might have eluded the authors. Second, an invitation to serve as a reviewer is an implicit acknowledgement of one's expertise and standing in the field and can be used as a proxy for these in consideration for academic advancement. Third, by reading numerous manuscripts at a formative stage, with varying levels of quality, one can learn quickly what does and does not work well in communicating one's science. Finally, although, of course, every unpublished manuscript must be considered a privileged communication, those who participate frequently in the peer review process have an organic opportunity to assimilate the latest developments and techniques in their chosen sub-specialization, and often beyond.

Beyond these intellectual and professional rewards, it is worth recognizing the truly substantial contribution that peer review represents to the scholarly enterprise. In many universities, peer review is not yet consistently or formally captured within workload allocation models, despite requiring considerable expertise, judgement and time. High-quality reviews have been estimated to take between 4 and 8 h to complete, depending on the complexity of the submission (Kovanis et al., 2016; Publons, 2018). When considered across the millions of reviews conducted globally each year, this represents an extraordinary collective investment by the academic community. Conservative estimates suggest that, when benchmarked against academic salaries, this contribution amounts to several billion US dollars annually (Nature Editorial, 2018; Research Information Network, 2008). Rather than exposing a weakness, this underscores the enduring strength of peer review as a system grounded in professionalism, collegiality and shared responsibility. As academic workloads continue to intensify, it also presents a timely opportunity for institutions to recognize and value peer review more explicitly as a core element of scholarly citizenship and research excellence. At the same time, we acknowledge that models of peer review continue to evolve, encompassing a spectrum of approaches that differ in transparency, structure and technological support. Although no single model is universally optimal, the shared objective remains constant: to preserve rigorous, fair and constructive evaluation that strengthens both individual manuscripts and the scientific record as a whole. Nevertheless, we agree with Gaudino et al. (2021) that additional investigation is warranted to identify mechanisms whereby peer review, so vital for the research enterprise overall and for the career progress of individual scientists, can be optimized further.

And despite the aforementioned benefits, it remains incumbent on editors, such as ourselves, to understand the importance of acknowledging the vital role that reviewers play in the publishing ecosystem and how valuable it becomes to identify individuals who can be relied upon to complete their duties robustly and promptly and who subscribe to the 'golden rule'. Crucially, editorial decision-making is not undertaken in isolation but represents a genuine partnership between editors and reviewers. Thoughtful, independent peer review provides the intellectual foundation upon which fair, balanced and proportionate editorial decisions are made, and it is this collaborative process that underpins the confidence placed in our journals by authors and readers alike. Indeed, peer review has been characterized in our pages as a defining feature that distinguishes science from non-science (Berg et al., 2024). Many hundreds of scientists contribute their insights to our editorial processes every year, and we are deeply indebted to them. For years, we have experimented with various mechanisms to reward our top peer reviewers from these groups, from baseball caps to T-shirts to notebooks, and have been encouraged by the positive reception that these gestures have received. However, we felt it was time to do more and to recognize excellence in peer review more visibly and more formally.

For this reason, we have launched our annual 'Exceptional Referees'

initiative for both *The Journal of Physiology* and *Experimental Physiology* [https://physoc.onlinelibrary.wiley.com/hub/exceptional-referees]. Here, we have published the first list of the colleagues who, amongst our myriad dedicated contributors, particularly stood out for the number, timeliness and exceptional quality of their reports in 2025. Their thoughtful evaluations, constructive feedback and commitment to scientific excellence have had a significant impact on our journals and the authors who publish within them.

Our Exceptional Referees across both journals represent a diverse snapshot of our field. Not surprisingly, they are concentrated in the USA and UK, where the majority of our submissions originate, in addition to other anglophone countries, such as Australia, New Zealand, Canada and Ireland. Importantly, however, there is also meaningful representation from non-English-speaking regions, including various European countries, Japan and Mexico. This underscores the truly international character of our editorial process. We were unfortunately unable to list a few individuals identified as Exceptional Referees (four each for *The Journal of Physiology* and *Experimental Physiology*) because they did not get back to us with their permission to be identified online. Notably, one female scientist appears on both lists, reflecting an exceptional level of engagement across journals.

The dually listed reviewer identified in the preceding paragraph is also one of only 14 listed referees (of 63) whom we are able to identify as female, which is <25% of the total Exceptional Referees. This prompts an important question: why does such a clear underrepresentation persist? Although it is well recognized that fewer female than male scientists are invited to review, there is also anecdotal evidence that prospective female reviewers might decline invitations at a higher rate. This might, in part, reflect the disproportionate service and student-facing responsibilities often carried alongside research, teaching and leadership roles, in addition to competing pressures on time.

Seen through this lens, peer review offers an important opportunity to reflect more broadly on equity and inclusion within the discipline. Evidence suggests that women and early-career researchers often carry substantial service responsibilities that lack formal recognition, while balancing competing demands on their time (Helmer

et al., 2017; Lerback Hanson, 2017). Ensuring that peer review activity is visible, valued and appropriately acknowledged, therefore, offers a positive route to strengthening fairness, retention and career development across the discipline. As editors, we see it as part of our role to champion the recognition of peer review within institutions, funding bodies and promotion frameworks. Although journals can celebrate and reward excellence directly, long-term sustainability will be achieved best when universities and employers explicitly recognize peer review as essential scholarly work that underpins research integrity, quality assurance and public trust in science. We have made great strides in increasing the gender balance of our editorial boards and senior editorial teams, and it would be great to see this reflected also in our Exceptional Referees. If you are a female physiologist, we therefore hope you will consider accepting future invitations to review, knowing that this contribution is both valued and visible.

In closing, therefore, we owe a profound debt of gratitude not only to our Exceptional Referees, but also to the many hundreds of reviewers and editorial board members who give so much of their time and energy to elevate the standing of physiological research. We are also grateful to mentors who take it upon themselves to involve their trainees and other early-career researchers in the peer review process, either by inviting them to serve as co-reviewers under their guidance or by recommending them for independent assignments. This vast, self-renewing scholarly community safeguards the quality, novelty and integrity of the work we publish. Quite literally, we couldn't do it without you!

Kim E. Barrett, Editor in Chief, *The Journal of Physiology*

University of California Davis School of Medicine

Damian M. Bailey, Editor in Chief, *Experimental Physiology*

University of South Wales

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

## Additional information

### Competing interests

The authors declare they have no competing interests.

### Author contributions

Kim E. Barrett and Damian M. Bailey both participated in drafting the manuscript and in editing the draft for critical intellectual content both initially and following peer review.

### Funding

None.

### Acknowledgements

This article has been simultaneously co-published by *The Journal of Physiology* and *Experimental Physiology*. The articles are identical except for minor stylistic and spelling differences in keeping with each journal's style. Either citation can be used when citing this article.

## Supporting information

Additional supporting information can be found online in the Supporting Information section at the end of the HTML view of the article. Supporting information files available:

**Peer Review History**

