## [Peer Review History · The Journal of Physiology]

Recognizing Excellence in Peer Review

Kim E Barrett and Damian Miles Bailey

DOI: 10.1113/JP290779

Corresponding author(s): Kim Barrett (kbarrett@ucdavis.edu)

Review Timeline:	Submission Date:	09-Feb-2026
	Editorial Decision:	23-Feb-2026
	Revision Received:	24-Feb-2026
	Accepted:	25-Feb-2026

Senior Editor: Laura Bennet

Reviewing Editor: Peking Fong

Transaction Report:

Dear Kim,

Re: JP-E-2026-290779 "The best of the best" by Kim E Barrett and Damian Miles Bailey

Thank you for submitting your manuscript to The Journal of Physiology. It has been assessed by a Reviewing Editor and by 1 expert referee and we are pleased to tell you that it is acceptable for publication following satisfactory revision.

The review comments are copied at the end of this email.

Please address all the points raised and incorporate all requested revisions or explain in your Response to Referees why a change has not been made. We hope you will find the comments helpful and that you will be able to return your revised manuscript within 2 weeks. If you require longer than this, please contact journal staff: jp@physoc.org.

REVISION CHECKLIST:

We look forward to receiving your revised submission.

Best wishes,

Laura Bennet
Senior Editor
The Journal of Physiology

EDITOR COMMENTS

Reviewing Editor:

Thank you for this timely and important editorial.

The Referee's comments are provided below. To summarize, the Referee identifies potential opportunities to incorporate insights published in both *The Journal of Physiology* and *Experimental Physiology*, as well as those from a study in *The Journal of the American Heart Association*.

In addition to the Referee's suggestions, I did make a few minor notes for your consideration:

1. Consider placing the statements starting at line 82 and ending at line 84 at the start of the next paragraph. Perhaps this would make the topical transition smoother: i.e. by posing a reason to probe data about the observation (low acceptance rates by female scientists)?
2. I remain a bit mystified by the choice of the title; it seems cryptic. However, a strong alternative eludes me at this time.

REFEREE COMMENTS

Referee #1:

This editorial by Barrett and Bailey, Editors-in-Chief of *The Journal of Physiology* and *Experimental Physiology*, respectively, thoughtfully reflects on the benefits of peer review. It appropriately highlights the altruistic contribution of reviewers. The piece brings to mind the earlier editorial by Forsythe, *Celebrating the quality of our referees* (*J Physiol.* 2017;595:6369-6370), which similarly acknowledged the essential contributions of reviewers. Perhaps a brief nod to this previous article may provide useful continuity and reinforce the two journals' recognition of reviewer efforts?

In addition, a recent co-published editorial (Berg et al., *Peer review: the imprimatur of scientific publication*, *Journal of Physiology*, 2024;602:4079-4083, *Experimental Physiology*, 2024;109:1407-1411;), discusses the historical foundations and scientific impact of peer review. It characterises peer review as one of the defining features distinguishing science from non-science. It may be worth briefly referencing this perspective in the introductory paragraph. The authors may also wish to consider the relatively recent systematic review and meta-analysis by Gaudino et al. (*Journal of the American Heart Association*, 2021;10:e019903), which examined experimental interventions aimed at improving the biomedical peer-review process. This study reported that reviewer-level interventions were associated with improved review quality, although often at the cost of increased review duration; reviewer blinding was a notable exception. In contrast, author- and editor-level interventions did not significantly affect either review quality or process duration. Please consider above as kind and constructive suggestions only. Overall, this is a timely and well-considered editorial.

END OF COMMENTS

Response to editor and reviewer

EDITOR COMMENTS

Reviewing Editor:

Thank you for this timely and important editorial.

Response: Thank you for your kind assessment.

The Referee's comments are provided below. To summarize, the Referee identifies potential opportunities to incorporate insights published in both The Journal of Physiology and Experimental Physiology, as well as those from a study in The Journal of the American Heart Association.

Response: We agree that these suggestions are most helpful. We have edited the manuscript to now refer to all of the articles suggested.

In addition to the Referee's suggestions, I did make a few minor notes for your consideration:

1. Consider placing the statements starting at line 82 and ending at line 84 at the start of the next paragraph. Perhaps this would make the topical transition smoother: i.e. by posing a reason to probe data about the observation (low acceptance rates by female scientists)?

Response: Thank you for suggesting this. We agree that this is a preferable transition and have edited the manuscript accordingly.

2. I remain a bit mystified by the choice of the title; it seems cryptic. However, at a strong alternative eludes me at this time.

Response: The corresponding author is obviously mistaken that this is a well-known idiom. We have revised the title to better advertise the contents of the editorial.

REFEREE COMMENTS

Referee #1:

This editorial by Barrett and Bailey, Editors-in-Chief of The Journal of Physiology and Experimental Physiology, respectively, thoughtfully reflects on the benefits of peer review. It appropriately highlights the altruistic contribution of reviewers. The piece brings to mind the earlier editorial by Forsythe, Celebrating the quality of our referees (J Physiol. 2017;595:6369-6370), which similarly acknowledged the essential contributions of reviewers. Perhaps a brief nod to this previous article may provide useful continuity and reinforce the two journals' recognition of reviewer efforts?

Response: We agree with this helpful suggestion and have now cited this prior editorial.

In addition, a recent co-published editorial (Berg et al., Peer review: the imprimatur of scientific publication, Journal of Physiology, 2024;602:4079-4083, Experimental Physiology, 2024;109:1407-1411;), discusses the historical foundations and scientific impact of peer review. It characterises peer review as one of the defining features distinguishing science from non-science. It may be worth briefly referencing this perspective in the introductory paragraph.

Response: We agree and have now cited this co-published editorial and this important concept.

The authors may also wish to consider the relatively recent systematic review and meta-analysis by Gaudino et al. (Journal of the American Heart Association, 2021;10:e019903), which examined experimental interventions aimed at improving the biomedical peer-review process. This study reported that reviewer-level interventions were associated with improved review quality, although often at the cost of increased review duration; reviewer blinding was a notable exception. In contrast, author- and editor-level interventions did not significantly affect either review quality or process duration.

Response: Thank you for alerting us to this interesting analysis, which has now been discussed briefly and appropriately cited.

Please consider above as kind and constructive suggestions only. Overall, this is a timely and well-considered editorial.

Response: Thank you for your kind words and constructive suggestions.

Dear Kim,

Re: JP-E-2026-290779R1 "Recognizing Excellence in Peer Review" by Kim E Barrett and Damian Miles Bailey

Thank you for your revisions and thank you for an excellent editorial.

We are pleased to tell you that your paper has been accepted for publication in The Journal of Physiology.

TRANSPARENT PEER REVIEW POLICY: To improve the transparency of its peer review process, The Journal of Physiology publishes online the peer review history of all articles accepted for publication as supporting information. Readers will have access to decision letters, including Editors' comments and referee reports, for each version of the manuscript, as well as any author responses to peer review comments. Referees can decide whether or not they wish to be named on the peer review history document.

Authors should note that it is too late at this point to offer corrections prior to proofing. The accepted version will be published online, ahead of the copy edited and typeset version being made available. Major corrections at proof stage, such as changes to figures, will be referred to the Editors for approval before they can be incorporated. Only minor changes, such as to style and consistency, should be made at proof stage. Changes that need to be made after proof stage will usually require a formal correction notice.

Best wishes,

Laura Bennet
Senior Editor
The Journal of Physiology

P.S. - You can help your research get the attention it deserves! Check out Wiley's free Promotion Guide for best-practice recommendations for promoting your work at www.wileyauthors.com/eoo/guide. You can learn more about Wiley Editing Services which offers professional video, design, and writing services to create shareable video abstracts, infographics, conference posters, lay summaries, and research news stories for your research at www.wileyauthors.com/eoo/promotion.

• **IMPORTANT NOTICE ABOUT OPEN ACCESS:** To assist authors whose funding agencies mandate immediate public access to published research findings, The Journal of Physiology allows authors to pay an Open Access (OA) fee to have their papers made freely available immediately on publication.

The Corresponding Author will receive an email from Wiley with details on how to register or log in to Wiley Authors where you will be able to place an order.

You can check if your funder or institution has a Wiley Open Access Account here:
<https://authors.wiley.com/author-resources/Journal-Authors/open-access/author-compliance-tool.html>